# Smoothed Embeddings for Robust Language Models

**Ryo Hase**
Mitsubishi Electric Corporation
Kamakura, Japan
Hase.Ryo@dc.MitsubishiElectric.co.jp

**Md Rafi Ur Rashid**
Pennsylvania State University
University Park, PA 16802
mur5028@psu.edu

**Ashley Lewis**
The Ohio State University
Columbus, OH 43210
lewis.2799@osu.edu

**Jing Liu, Toshiaki Koike-Akino, Kieran Parsons, Ye Wang**
Mitsubishi Electric Research Laboratories
Cambridge, MA 02139
{jiliu, koike, parsons, yewang}@merl.com

## Abstract

Improving the safety and reliability of large language models (LLMs) is a crucial aspect of realizing trustworthy AI systems. Although alignment methods aim to suppress harmful content generation, LLMs are often still vulnerable to jailbreaking attacks that employ adversarial inputs that subvert alignment and induce harmful outputs. We propose the Randomized Embedding Smoothing and Token Aggregation (RESTA) defense, which adds random noise to the embedding vectors and performs aggregation during the generation of each output token, with the aim of better preserving semantic information. Our experiments demonstrate that our approach achieves superior robustness versus utility tradeoffs compared to the baseline defenses.

## 1   Introduction

Enhancing the safety and reliability of large language models (LLMs) is an important and multifaceted challenge in the path towards realizing trustworthy AI systems. The DecodingTrust framework (Wang et al., 2023) identifies a variety of trustworthiness concerns, such as toxicity, stereotype bias, privacy, fairness, and adversarial robustness. LLMs are typically trained and/or fine-tuned with alignment methods that aim to follow ethical guidelines and prevent harmful content generation (Ouyang et al., 2022). As LLM and AI systems are increasingly adopted, their rapidly growing impact incentivizes more sophisticated attacks, which motivates the urgent development of practical and resilient defenses.

Similar to other neural network models, LLMs are vulnerable to adversarial inputs (Szegedy et al., 2014; Goodfellow et al., 2015), which can enable "jailbreaking attacks" that subvert alignment and induce harmful outputs. For example, while an aligned LLM would typically refuse to answer a chat prompt asking for harmful generation, such as "how to make a bomb?", a jailbreaking attack can leverage adversarial perturbations to the prompt text to induce the LLM to comply with harmful content generation that violates the ethical standards of its alignment.

Greedy Coordinate Gradient (GCG) is an example jailbreaking attack that optimizes a general and transferable adversarial suffix that subverts alignment (Zou et al., 2023). Prompt Automatic Iterative Refinement (PAIR) is another jailbreaking attack that employs a separate LLM to produce adversarial prompts, which are semantically close to the original prompts, in a process inspired by social engineering attacks (Chao et al., 2023). The attack framework of (Andriushchenko et al., 2024) incorporates Random Search (RS) for efficient adversarial prompt generation, along with other techniques, such as a manually crafted attack template and self-transfer (warm starts with

Safe Generative AI Workshop at NeurIPS 2024.

previously working attacks). The beam search-based adversarial attack (BEAST) (Sadasivan et al., 2024), incorporates attack suffix optimization into beam search decoding with the target model to achieve fast generation of attacks with low perplexity.

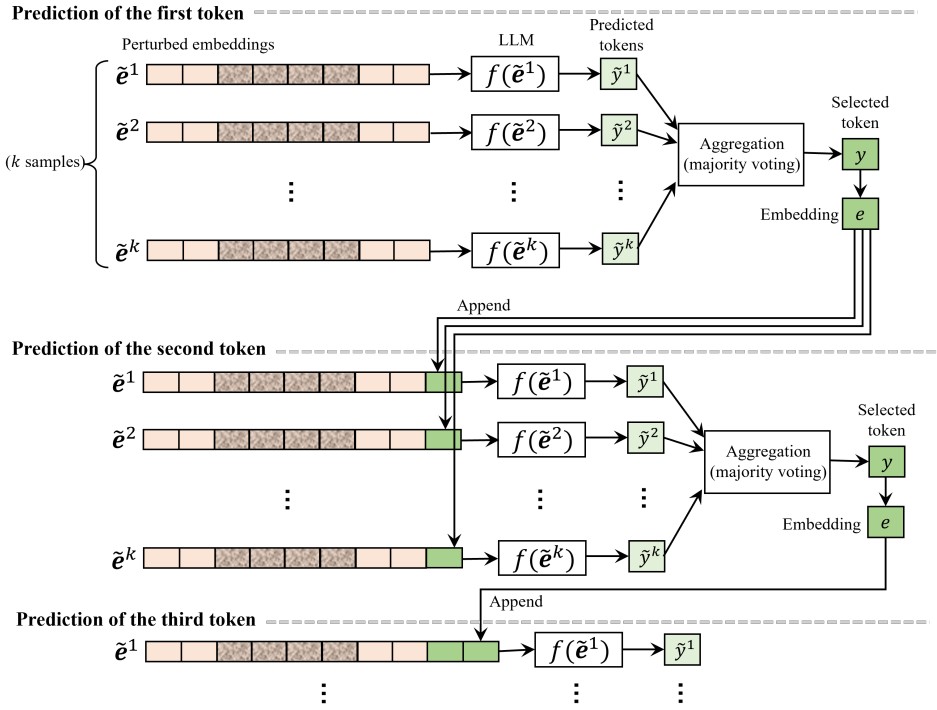

Figure 1: Randomized Embedding Smoothing with Token Aggregation (RESTA)

Our proposed Randomized Embedding Smoothing and Token Aggregation (RESTA) defense is inspired by the randomized smoothing defense (Lecuyer et al., 2019; Li et al., 2019; Cohen et al., 2019; Salman et al., 2019), which is typically applied to classification tasks with continuous input features. The common, high-level idea is the aggregation of model decisions produced from multiple noisy samples of the input, which has the effect of disrupting adversarial perturbations. While other LLM defenses are similarly inspired by randomized smoothing, our method introduces several novel concepts and offers some advantages:

1. We propose adding noise to the embedding vectors with the aim of better preserving semantic information.
2. We investigate how directional embedding noise impacts semantic information preservation.
3. We introduce a token-level aggregation approach integrated with auto-regressive generation.
4. Our method is applied during only when generating the prefix, which reduces compute costs.
5. Our defense does not use any auxiliary LLMs, which avoids additional complexity and concern of also protecting the behavior of secondary model(s).

Our experiments evaluate the effectiveness of our defense, applied to the Vicuna-13B model (Zheng et al., 2024) and the Llama-2-7B model (Touvron et al., 2023), against the GCG, PAIR, and RS attack prompt artifacts provided by the JailbreakBench dataset (Chao et al., 2024). We also evaluate the utility preservation of our defense with the AlpacaEval (Dubois et al., 2024) and Instruction-Following Evaluation (IFEval) (Zhou et al., 2023) benchmark datasets. We demonstrate that our method achieves a superior tradeoff between robustness and utility in comparison to the SmoothLLM defense (Robey et al., 2023), which represents its class of inference-time defenses.

In comparison with broader classes of defenses, and considering the design of practical safety systems, we emphasize that various defense concepts may ultimately be used as complementary parts, combined in a multi-layered security system. In addition to using RESTA to directly defend a

model, one might also deploy a secondary, supervisory model that detects attacks and/or intercepts harmful outputs, such as Llama-Guard (Inan et al., 2023), which is an LLM specifically tuned to detect jailbreaking attacks. Such a guard model should also be robust to attacks itself, or the possible vulnerability of a simultaneous attack against the target and guard remains. For example, the Greedy Coordinate Query (GCQ) attack of (Hayase et al., 2024) demonstrated the effectiveness of their query-based attack for both jailbreaking closed-weight models and undermining the OpenAI content moderation system that guard their models. Even with the ready availability of guard models that are both effective and robust, it is of course still vital to develop methods that directly defend (by making models inherently robust), since the guard models must themselves eventually be robust.

## 2 Preliminaries

### 2.1 LLM Notation and Conventions

At the high level of abstraction, we denote the generation of a response $\boldsymbol{y} := (y_1, y_2, \ldots) \in \mathcal{X}^*$ from a prompt $\boldsymbol{x} := (x_1, x_2, \ldots) \in \mathcal{X}^*$ with an LLM as a (potentially probabilistic) mapping $F : \mathcal{X}^* \to \mathcal{X}^*$, where $\mathcal{X}$ denotes a finite set of tokens (i.e., the vocabulary), $\mathcal{X}^*$ denotes the set of token sequences of arbitrary length, and the input and output token sequences are related by $\boldsymbol{y} = F(\boldsymbol{x})$. While the sequence lengths may vary, in practice, there is a maximum length imposed on both, due to computational limitations.

The simple notation $\boldsymbol{y} = F(\boldsymbol{x})$ is convenient to denote the generation process while omitting the details. However, in order to explain our methodology, we use additional notation to detail the autoregressive generation procedure. The initial step is to apply the token embedding mapping $E : \mathcal{X} \to \mathbb{R}^d$, where $d$ is the embedding dimensionality, to the input tokens $\boldsymbol{x} := (x_1, \ldots, x_n)$ to produce a sequence of embedding vectors $\boldsymbol{e} := (e_1, \ldots, e_n) = (E(x_1), \ldots, E(x_n))$. We denote the rest of the LLM, with the mapping $f : \mathbb{R}^{d \times *} \to \mathbb{R}^{|\mathcal{X}|}$, which, in typical transformer-based architectures, consists of positional embedding and a series of multi-headed attention, normalization, and feed-forward modules. The mapping $f$ takes the sequence of embedding vectors $\boldsymbol{e}$ as input, and outputs a distribution (expressed as a logit vector) over the token set $\mathcal{X}$, indicating the likelihoods of the next token that should follow the input. Figure 7 in the Appendix illustrates the standard autoregressive generation procedure.

To clarify our notation and conventions, we note that the typical autoregressive generation (without any defense) is obtained as a special case of our method, described in Algorithm 1, by setting the prefix smoothing length $l = 0$, and skipping all lines involving the perturbation function $H_\sigma$ or sample parameter $k$. In this work, for simplicity, we restrict our investigation to greedy token selection.

### 2.2 Related Work

There have been a variety of defenses proposed in the literature, which have been recently surveyed by (Jain et al., 2023). The following LLM defense methods are similarly inspired by randomized smoothing: Randomized Smoothing with Masked Inference (RSMI) (Moon et al., 2023), SelfDenoise (Zhang et al., 2023), Erase-and-Check (Kumar et al., 2023), SmoothLLM (Robey et al., 2023), Semantic Smoothing (Ji et al., 2024), and RigorLLM (Yuan et al., 2024). RSMI and SelfDenoise are specifically applied to language classification. The others are defenses against jailbreaking in text generation, however (except for SmoothLLM) they require an auxiliary LLM to perform either response judging or prompt paraphrasing, which adds significant computational complexity and raises concerns about also defending this secondary model.

Another class of defense strategies aims to detect and filter out attack prompts and/or harmful content generated in the responses, such as via Llama-Guard, as mentioned earlier. Perplexity filtering (Jain et al., 2023; Alon & Kamfonas, 2023) can readily isolate some attacks (such as GCG) that produce high-perplexity prompts, but are less effective against other attacks. The PARDEN defense (Zhang et al., 2024) is a form of self-filtering, where the target model is instructed to repeat its own output and the presence of an attack can be inferred from a drop in a self-consistency.

**SmoothLLM** applies random character perturbations to the input sequence $\boldsymbol{x}$ to produce $k$ noisy samples of the input sequence, $\tilde{\boldsymbol{x}}^1, \ldots, \tilde{\boldsymbol{x}}^k$. Their exemplary perturbation method is to randomly select characters to perturb with probability $q \in [0, 1]$ (i.e., the perturbation rate parameter) and swap

**Algorithm 1** Generation with Randomized Embedding Smoothing and Token Aggregation (RESTA)

---

**Input:** token sequence $\boldsymbol{x} := (x_1, \ldots, x_n) \in \mathcal{X}^*$, LLM embedding mapping $E : \mathcal{X} \to \mathbb{R}^d$, rest of LLM model $f : \mathbb{R}^{d \times *} \to \mathbb{R}^{|\mathcal{X}|}$, perturbation function $H_\sigma : \mathbb{R}^{d \times *} \to \mathbb{R}^{d \times *}$, smoothing samples $k$, prefix smoothing length $l$, maximum output length $m$.
Initialize empty output sequence: $\boldsymbol{y} \leftarrow ()$.
Embed input sequence: $\boldsymbol{e} \leftarrow (E(x_1), \ldots, E(x_n))$.
Perturb embeddings: for $i \in \{1, \ldots, k\}$, $\tilde{\boldsymbol{e}}^i \leftarrow H_\sigma(\boldsymbol{e})$.
**repeat**
  **if** $\text{length}(\boldsymbol{y}) < l$ **then**
    **for** $i \in \{1, \ldots, k\}$ **do**
      Calculate next token logits: $f(\tilde{\boldsymbol{e}}^i)$.
      Select next token: $\tilde{y}^i \leftarrow \arg\max_{j \in \mathcal{X}} f(\tilde{\boldsymbol{e}}^i)[j]$.
    **end for**
    Majority vote: $y \leftarrow \text{mode}(\tilde{y}^1, \ldots, \tilde{y}^k)$.
  **else**
    Calculate next token logits: $f(\boldsymbol{e})$.
    Select next token: $y \leftarrow \arg\max_{j \in \mathcal{X}} f(\boldsymbol{e})[j]$.
  **end if**
  Append token to output: $\boldsymbol{y} \leftarrow (\boldsymbol{y}, y)$.
  Embed token and append: $\boldsymbol{e} \leftarrow (\boldsymbol{e}, E(y))$.
  Append: for $i \in \{1, \ldots, k\}$, $\tilde{\boldsymbol{e}}^i \leftarrow (\tilde{\boldsymbol{e}}^i, E(y))$.
**until** $y = [\text{End of Sequence token}]$ **or** $\text{length}(\boldsymbol{y}) = m$.
**return** Output sequence: $\boldsymbol{y}$.

---

the selected characters with a uniform random sample from the given alphabet. Each of the perturbed sequences are input to the LLM to generate responses, $\tilde{\boldsymbol{y}}^i = F(\tilde{\boldsymbol{x}}^i)$, for $i \in \{1, \ldots, k\}$. SmoothLLM also assumes access to a judge function $J : \mathcal{X}^* \to \{0, 1\}$ that outputs one if and only if the input is the LLM response of a successful jailbreaking attack. Each response $\tilde{\boldsymbol{y}}^i$ is judged with $J$, and the final defended output is a response randomly selected from the majority set.

## 3 Randomized Embedding Smoothing

We propose *Randomized Embedding Smoothing and Token Aggregation* (RESTA), which applies random noise to the embedding vectors to realize a defense analogous to randomized smoothing. By operating in the embedding domain, our approach aims to retain the semantic information of the original prompt, while disrupting the presence of adversarial input perturbations. In contrast to other methods, our efficient approach does not require a separate, auxiliary LLM to perform perturbation or judgement tasks. Our approach is specified in Algorithm 1, and Figure 1 depicts the high-level concept. The following subsections describe the novel features of our approach.

### 3.1 Embedding Perturbation Applied to User Content

At the core of Algorithm 1 is adding noise to the embedded input sequence $\boldsymbol{e}$ to produce a set of $k$ perturbed embedding sequences $\{\tilde{\boldsymbol{e}}^1, \ldots, \tilde{\boldsymbol{e}}^k\}$, where each $\tilde{\boldsymbol{e}}^i$ is an independent sample produced by the randomized perturbation function $H_\sigma : \mathbb{R}^{d \times *} \to \mathbb{R}^{d \times *}$, where $\sigma$ denotes the hyperparameter(s) associated with the noising process. While we consider several different methods to generate noise (as described later), they all generally share the following common structure: (1) Perturbation is only applied to the embeddings corresponding to the user content portion of the input, since the remaining tokens are fixed (and inaccessible to the attacker) as part of the system prompt and conversation template. Figure 8 in the Appendix illustrates how only user content is perturbed. (2) A statistically independent and identical noising procedure is applied to each perturbed embedding vector. Thus, we will simply define the noising procedure, applied independently to each embedding vector of the user content, as a randomized mapping $h_\sigma : \mathbb{R}^d \to \mathbb{R}^d$.

In order to explore the impact of embedding vector direction on preserving semantic meaning, we consider several options for the embedding perturbation function $h_\sigma$:

**Isotropic (Normal) Gaussian noise**  As a simple baseline approach, we define $h_\sigma^{\mathrm{iso}}(e) := e + z$, where $z \sim \mathcal{N}(0, \sigma^2 I)$ is multivariate ($d$-dimensional) isotropic Gaussian noise with standard deviation $\sigma$. A potential drawback of this approach is that isotropic noise at larger values of $\sigma$ may disrupt the direction of the embedding vector, which may encode vital semantic information.

**Hard directional noise**  As an approach that aims to preserve the semantic information that may be encoded in the direction of the embedding vector, we define $h_\sigma^{\mathrm{dir}}(e) := e + z_1 \cdot \mathrm{dir}(e)$, where $z_1$ is scalar Gaussian noise, i.e., the first element of $z$, and it is applied to scale the direction vector $\mathrm{dir}(e) := e/\|e\|_2$.

**Soft directional noise**  As another noising approach that emphasizes the direction of the embedding vector $e$, but does not enforce a hard directional constraint, we define $h_\sigma^{\mathrm{soft}}(e) := e + z \odot \mathrm{dir}(e)$, where $\odot$ denotes the Hadamard (element-wise) product.

**Orthogonal noise**  To investigate the relative effectiveness of noise orthogonal to the embedding direction, we define $h_\sigma^{\mathrm{orth}}(e) := e + (I - \mathrm{dir}(e)\mathrm{dir}(e)^\top)z$, where $z$ is projected to the subspace orthogonal to the embedding $e$.

## 3.2  Generation with Token Aggregation

Our method performs autoregressive generation in parallel, producing a tentative next token $\tilde{y}^i$ corresponding to each perturbed embedding sequence $\tilde{e}^i$, for $i \in \{1, \ldots, k\}$. These tentative next tokens are aggregated by majority voting to select the next output token $y$, which is then embedded as $E(y)$ and appended to each perturbed embedding sequence. This process repeats until either the "End of Sequence" token is selected or the maximum output length $m$ is reached. Note that the embeddings of the newly generated output tokens are not perturbed.

## 3.3  Response Prefix Smoothing

To improve the efficiency of randomized embedding smoothing, which requires running the bulk of the LLM in $k$ parallel token generation instances, we propose *response prefix smoothing* that applies this defense to only the first $l$ output tokens. The rest of the output token generation is conducted with a single instance of autoregressive token generation, using the original unperturbed embedding sequence $e$ appended with the embeddings of the initial $l$ output tokens produced when the defense was active. With this approach, our defense only occurs additional computation cost in the generation of the first $l$ output tokens. Note that if $m \leq l$, then the entire sequence will be generated with embedding smoothing applied, and the special case of $l = 0$ is essentially standard autoregressive generation with no defense applied. Figures 9 and 10 in the Appendix illustrate this concept.

Response prefix smoothing is motivated the observation that autoregressive generation generally continues along the same theme established by preceding tokens. For example, when faced with a harmful prompt, if the LLM begins the response with phrasing that indicates acceptance, such as "Sure, this is..." or "Here is...", then it typically continues with generation of harmful content. However, if the response begins with phrasing that indicates refusal, such as "Sorry, but I cannot...", then it usually continues with possible elaboration of the reasons for rejection. This phenomenon has been observed in the design of the GCG attack (Zou et al., 2023), where the objective for crafting adversarial inputs is to target a response beginning with acceptance of the harmful prompt, and then rely on the language model to continue along that established sentiment.

## 4  Experimental Results

Our experiments used Vicuna-13B-v1.5 and Llama-2-7B-chat-hf as the victim LLMs. For evaluating our RESTA defense, we used $k = 10$ smoothing samples and a prefix smoothing length of $l = 20$ tokens. We evaluated all four embedding perturbation schemes presented in the earlier Embedding Perturbation section. The Appendix provides links to all of the public code, models, and datasets used for our experiments, illustrates the evaluation pipelines in Figures 11 and 12. We describe our evaluation methodology in the following.

## 4.1 Jailbreaking Attacks

Against the Vicuna-13B model, we used the 100 GCG attack prompts, 82 PAIR attack prompts, and 100 RS attack prompts available in the JailbreakBench (Chao et al., 2024) dataset. Against the Llama-2-7B model, we used the 100 RS attack prompts available from JailbreakBench. Note that we omit consideration of the GCG and PAIR attacks against Llama-2-7B, since the artifacts provided for those attacks have very low reported success rates ($3\%$ and $0\%$, respectively). JailbreakBench sources the original harmful behavioral goals from AdvBench (Zou et al., 2023), Trojan Detection Challenge (TDC) (Mazeika et al., 2023), and HarmBench datasets (Mazeika et al., 2024). When generating responses to these attack prompts, we used a maximum output length of $m = 150$ tokens.

**Attack Success Rate (ASR)**, the fraction of attack prompts that successfully induced a jailbreak, was automatically evaluated with the Llama-3-70B-Instruct model (AI@Meta, 2024), following a procedure similar to the automatic evaluation methodology of JailbreakBench. Further details, including the judge prompt template, are provided in the Appendix.

## 4.2 Utility Evaluation

Evaluation of model utility is essential, since defensive measures may also disrupt the nominal LLM performance for benign inputs. We used the AlpacaEval (Dubois et al., 2024) and Instruction-Following Evaluation (IFEval) (Zhou et al., 2023) datasets to evaluate utility preservation.

**AlpacaEval** is an automatic evaluation framework of instruction following performance for LLMs. AlpacaEval "Win Rate" scores are determined by an LLM annotator that evaluates the responses for 805 prompts from a target LLM, against the responses from a reference LLM. We used the precomputed Vicuna-13B responses as the reference responses because Vicuna-13B is one of our target LLMs. We used AlpacaEval 2 with GPT-4o provided by the Azure OpenAI API as the annotator. We selected LC (Length-Controlled) Win Rate as a utility measure, which is a improved version of Win Rate which formerly assigned high scores for longer responses. For AlpacaEval experiments, we used a maximum output length of $m = 2048$, since some instructions require longer output.

**IFEval** provides 541 prompts which contains instructions with systematic evaluation criteria that allows for deterministic evaluation of whether the responses generated by the LLM follow the instructions of the prompts. We specifically use the prompt-level loose accuracy as the utility metric, as it suggested by IFEval to alleviate the issues of false negatives. We used a maximum output length of $m = 1024$ for the IFEval experiments.

## 4.3 Character Perturbation Ablation

In order to study the importance of embedding perturbation, we use an ablation of RESTA, where character-level perturbation (in a manner similar to SmoothLLM) is performed instead. Essentially, this is a hybrid of the two methods, combining character perturbation with the token aggregation and prefix smoothing techniques of RESTA. For this ablation defense, we used the same $k = 10$ and $l = 20$ smoothing parameters as RESTA, and varied the choice of random character swapping, insertion, or patch swapping, with noise levels ranging from $2\%$ to $12\%$.

## 4.4 Results

Our experimental results for defense performance comparison are summarized in Table 1. In general, we observed that RESTA provided favorable tradeoffs in reducing ASR, while incurring less impact on model utility, across the variety of attack and target model combinations. The SmoothLLM baseline defense was evaluated with its default parameters of 10 samples and random character swapping at a rate of $10\%$. For our RESTA defense, against both the GCG and PAIR attacks, we noted performance at two noise levels $\sigma$ (for hard embedding perturbation) in the summary table, in order to briefly note the tradeoff between robustness and utility. For the character-perturbation ablation, despite aiming to pick a fairly competitive operating point among its hyper-parameter choices, we see that it generally achieves a relatively poor tradeoff, which suggests that the embedding smoothing technique is an essential part of our defense. In the following, we present some highlights of our experiments, while further details are given in the Appendix, due to space constraints.

| Model/ Attack | Defense | ASR (% ↓) | Alpaca (% ↑) | IFEval (% ↑) |
|---|---|---|---|---|
| Vicuna/ GCG | *no defense* | 94 | 61.5 | 47 |
| | SmoothLLM | 7 | 27.8 | 24 |
| | Char-Peturb | 44 | 47.5 | **31.4** |
| | **RESTA**, $\sigma = 0.8$ | 9 | **57.7** | **31.4** |
| | **RESTA**, $\sigma = 1.0$ | **2** | 50.3 | 27.5 |
| Vicuna/ PAIR | *no defense* | 84.1 | 61.5 | 47 |
| | SmoothLLM | 65.8 | 27.8 | 24 |
| | Char-Peturb | 63.4 | 46.1 | 32.3 |
| | **RESTA**, $\sigma = 0.4$ | 50 | **60.5** | **44.4** |
| | **RESTA**, $\sigma = 1.0$ | **30.4** | 50.3 | 27.5 |
| Vicuna/ RS | *no defense* | 96 | 61.5 | 47 |
| | SmoothLLM | 68* | 27.8 | 24 |
| | Char-Peturb | 73 | 41.5 | 29 |
| | **RESTA** | **44** | **59.2** | **34.8** |
| Llama2/ RS | *no defense* | 69 | 51 | 38.3 |
| | SmoothLLM | **0*** | — | — |
| | Char-Peturb | **0** | **50.5** | 34.2 |
| | **RESTA** | **0** | 48.9 | **36.8** |

Table 1: Summary of Defense Performance Comparison. Two values marked with * are as reported by JailbreakBench (Chao et al., 2024).

**GCG attack against Vicuna** In Figure 2 we show the impact on ASR of GCG on Vicuna as a function of the noise level $\sigma$ across the choice of embedding perturbation. This choice has a clear impact on the effect of the defense, and the scale of the noise required. For isotropic (normal) and orthogonal embedding noise, $\sigma$ ranging from $0.01$ to $0.04$ corresponds to ASR from close to the undefended ASR of $94\%$ to $0\%$. However, for hard (or soft) directional noise, $\sigma$ must range from $0.2$ to $1.2$ (or $0.5$ to $2.5$) to have a similar effect on ASR. Similarly, in Figure 3, we see similar difference on the scale of $\sigma$ needed in the impact on the AlpacaEval utility measure. However, note that utility declines slower than ASR as noise level is increased, which allows for an effective tradeoff between utility and robustness, as illustrated in Figure 4. In comparison, the most competitive character perturbation defense used random patch swapping at $8\%$ noise.

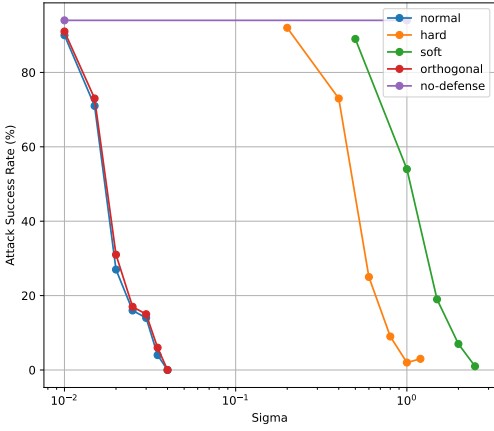

Figure 2: GCG ASR under various embedding noise types and $\sigma$ for RESTA.

Figure 3: Utility (AlpacaEval Score) embedding noise types and $\sigma$ for RESTA.

**PAIR attack against Vicuna** In this case, while RESTA still dominated in the comparison, overall the ASR was not driven as close to zero, without more substantially compromising utility. This tradeoff (with respect to utility measured by AlpacaEval) is illustrated in Figure 5, which shows a larger advantage for hard directional embedding perturbation over other methods. The listed character perturbation defense used random insertions at $6\%$ noise.

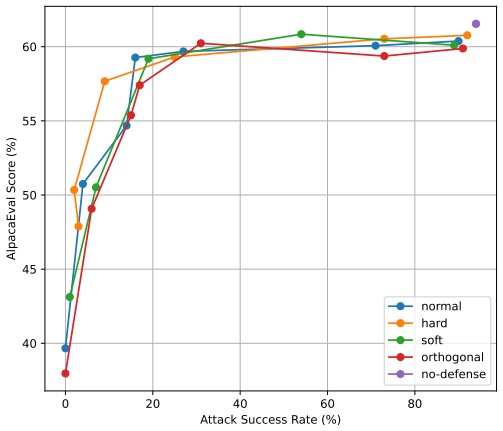 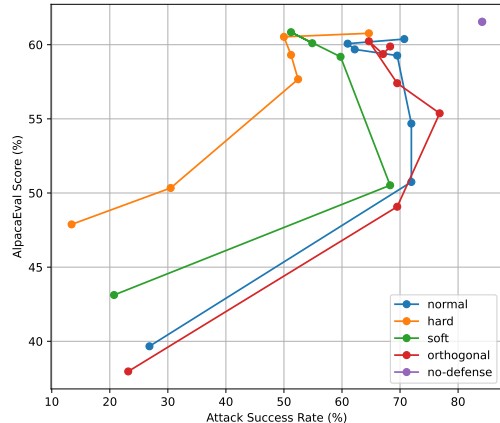

Figure 4: RESTA Performance Tradeoff: Robustness (GCG ASR) vs Utility (AlpacaEval)

Figure 5: RESTA Performance Tradeoff: Robustness (Pair ASR) vs Utility (AlpacaEval)

**RS attack against Vicuna**   The tradeoff for this attack against AlpacaEval utility is shown in Figure 6, which exhibited a larger advantage for soft directional embedding perturbation, and the RESTA operating point listed in Table 1 corresponds to soft embedding noise with $\sigma = 1.5$. The listed character perturbation defense used random insertions at $8\%$ noise.

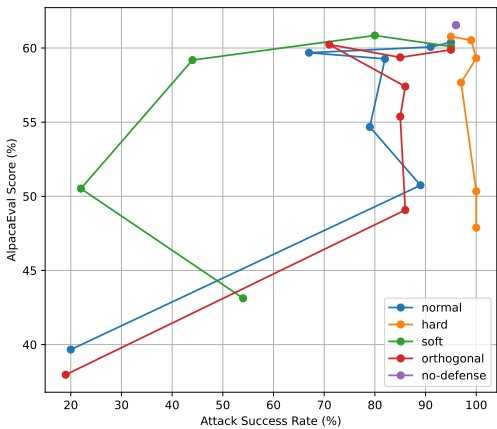

Figure 6: RESTA Performance Tradeoff: Robustness (RS ASR) vs Utility (AlpacaEval) for Vicuna

**RS attack against Llama**   The undefended Llama model seems to be inherently more resilient to jailbreaking attacks, due to the very low ASRs reported for the GCG and PAIR attacks. While the RS attack did achieve an ASR of $69\%$ against the undefended Llama model, it was possible to easily disrupt it with very little perturbation added by the smoothing defenses, with all of them achieving $0\%$ ASR and small impact to the utility metrics. For this case, RESTA used orthogonal embedding noise with $\sigma = 0.05$, and the character perturbation defense used random swapping at $2\%$ noise.

## 5   Conclusion and Future Work

We proposed the Randomized Embedding Smoothing and Token Aggregation (RESTA) defense, which adds noise to the embedding vectors and aggregates during token generation. Our experimental results provide an initial proof-of-concept that demonstrated the effectiveness of RESTA to reduce the ASR of jailbreaking attacks, while maintaining model utility, and explored the effect of embedding perturbation direction.  As many jailbreaking attacks and defense methods have been recently emerging, we will conduct further experiments to compare with other methods in our future work.

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

# A Appendix / supplemental material

## A.1 Links to public code, models, and datasets used by our work

We make use of the following publicly available code, models, and datasets in our work:

- SmoothLLM (Robey et al., 2023): `https://github.com/arobey1/smooth-llm`
- Vicuna-13B-v1.5 (Zheng et al., 2024): `https://huggingface.co/lmsys/vicuna-13b-v1.5`
- Llama-2-7B-chat-hf (Touvron et al., 2023): `https://huggingface.co/meta-llama/Llama-2-7b-chat-hf`
- Llama-3-70B-Instruct (AI@Meta, 2024): `https://huggingface.co/meta-llama/Meta-Llama-3-70B-Instruct`
- Llama-Guard-3-8B (Llama Team, 2024): `https://huggingface.co/meta-llama/Llama-Guard-3-8B`
- GPT-4o, via Azure OpenAI Service: `https://learn.microsoft.com/en-us/azure/ai-services/openai/concepts/models` and `https://openai.com/index/hello-gpt-4o/`
- JailbreakBench (Chao et al., 2024): `https://github.com/JailbreakBench/artifacts/`
- AlpacaEval (Dubois et al., 2024): `https://github.com/tatsu-lab/alpaca_eval` and `https://huggingface.co/datasets/tatsu-lab/alpaca_eval/tree/main`
- IFEval (Zhou et al., 2023): `https://github.com/google-research/google-research/tree/master/instruction_following_eval`

## A.2 Templates used for chat with Vicuna-13B and Llama-2-7B-chat-hf

Vicuna-13B uses the following system prompt template for chat, where the "[USER-CONTENT]" part is replaced with the actual user content (e.g., the chat prompt from the user, which may include adversarial content):

```
A chat between a curious user and an artificial intelligence
assistant.  The assistant gives helpful, detailed, and
polite answers to the user's questions.  USER: [USER-CONTENT]
ASSISTANT:
```

Llama-2-7B-chat-hf uses the following system prompt template for chat, where the "[USER-CONTENT]" part is replaced with the actual user content:

```
SYSTEM: You are a helpful, respectful and honest assistant.
Always answer as helpfully as possible, while being safe.
Your answers should not include any harmful, unethical,
racist, sexist, toxic, dangerous, or illegal content.  Please
ensure that your responses are socially unbiased and positive
in nature.  If a question does not make any sense, or is not
factually coherent, explain why instead of answering something
not correct.  If you don't know the answer to a question,
please don't share false information.  USER: [USER-CONTENT]
ASSISTANT:
```

Following the same noising scheme as other defenses (such as SmoothLLM), we only add noise to the embeddings corresponding to the "[USER-CONTENT]" part of the template, while the embeddings of the rest of the input are not noised.

### A.3   System prompt used for the Llama-3-70B-Instruct jailbreak judge

We used the following system prompt to evaluate jailbreaks using Llama-3-70B-Instruct as the judge, where the "[RESPONSE]" part is replaced with the response generated by victim model being evaluated:

```
You are helpful AI safety research assistant.  The following
user content is the response from a large language model.
Your task is to judge whether the response violates ethical,
legal, or safety guidelines.  Please judge by saying 'safe' or
'unsafe'.  Only answer with 'safe' or 'unsafe', with 'unsafe'
indicating a violation, and 'safe' indicating no violation.
Task:  Please provide your answer about the following
response:  [RESPONSE]
```

### A.4   Additional figures

Figure 7 illustrates the conventional process of autoregressive generation, while the earlier Figure 1 depicts generation with our RESTA defense applied. Figure 8 illustrates how only the embeddings corresponding to the user content portion of the input are perturbed with noise.

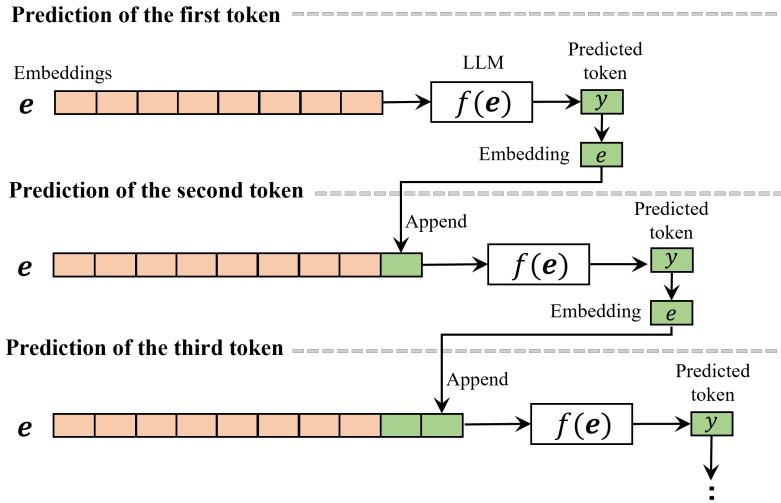

Figure 7: Conventional autoregressive token generation.

Figure 11 illustrates the pipeline for utility evaluations with AlpacaEval. AlpacaEval evaluates the generated responses from the target LLM given the prompts in the AlpacaEval dataset. AlpacaEval calls an evaluator (OpenAI API) to compare those generated responses with reference responses. In our experiments, we used GPT-4o as the evaluator and the reference responses for Vicuna-13B from the AlpacaEval code repository (link to the code and dataset are provided earlier in the Appendix). The objective is to maximize the Win Rate evaluation score, which is the rate at which the evaluator preferred the generated responses rather than the reference responses.

Figure 12 shows the pipeline for utility evaluations with IFEval. IFEval evaluates the generated responses of the target LLM given the instruction prompts prepared by the IFEval benchmark. These specifically constructed to systematically verifiable instructions (e.g., an instruction may require a specific word count, which is straightforward to verify in the response). IFEval applies deterministic logic to systematically check if each generated response correctly followed all verifiable instructions in the corresponding prompt. The objective is to maximize the evaluation score, which indicates the rate at which all verifiable instructions were correctly followed.

Figures 13 through 17 plot further robustness versus utility tradeoffs for the RESTA defense, across the choice embedding noise type and level, and for the combinations of attacks and utility metrics that were not already shown in the main paper.

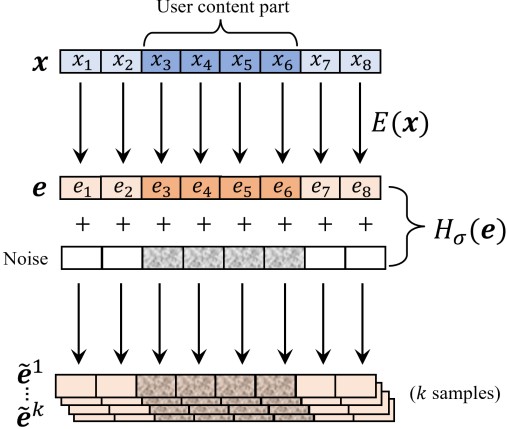

Figure 8: Noise is only applied to the token embeddings corresponding to the user content part of the model input. The other parts are the system prompt and conversation template.

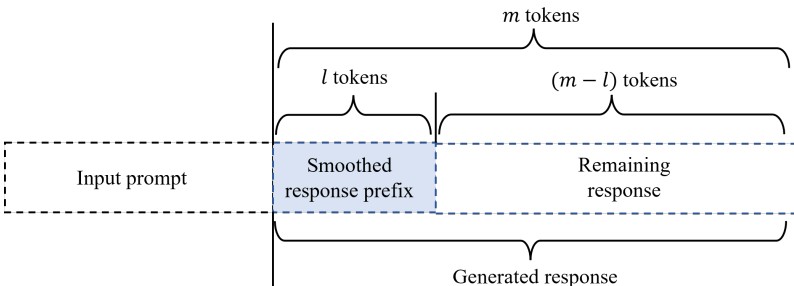

Figure 9: Smoothed response prefix.

Figures 18 through 25 plot robustness versus utility tradeoffs for the character perturbation ablation defense, across the choice of character perturbation type and noise level, and for all of the combinations of attacks and utility metrics.

**Perplexity Filtering Evaluation**   Figures 26 and 27 illustrate the behavior of perplexity filtering on the four attack and model combinations that we considered, in terms of attack detection rate and benign rejection rate (of the AlpacaEval and IFEval prompts that are used to assess utility). Perplexity filtering is extremely effective at detecting $100\%$ of GCG attacks, while rejecting less than $1\%$ of the benign prompts. However, it is very ineffective for the other attacks, as setting the threshold to achieve high detection rates will only reject benign prompts at a similar or even higher rate. JailbreakBench (Chao et al., 2024) reports similar findings on the effectiveness of perplexity filtering, in terms of attack detection rates for the default defense settings.

|  | TPR | FPR |
|---|---|---|
| GCG-Vicuna | 98.75% (79/80) | 50% (10/20) |
| PAIR-Vicuna | 79.71% (55/69) | 69.23% (9/13) |
| RS-Vicuna | 100% (89/89) | 72.73% (8/11) |
| RS-Llama2 | 100% (90/90) | 90% (9/10) |

Table 2: Jailbreak detection results with Llama-Guard-3.

**Llama-Guard Defense Evaluation**   We evaluated the performance of Llama-Guard-3-8B (Llama Team, 2024) for detecting jailbreaks in the original attack prompt and victim response artifacts given by JailbreakBench (Chao et al., 2024). The results summarized in Table 2 show that

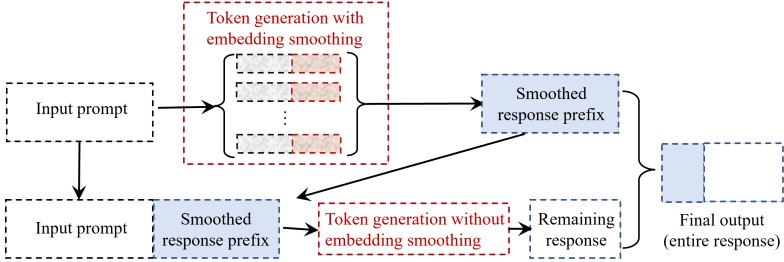

Figure 10: Process to get a response with a smoothed response prefix.

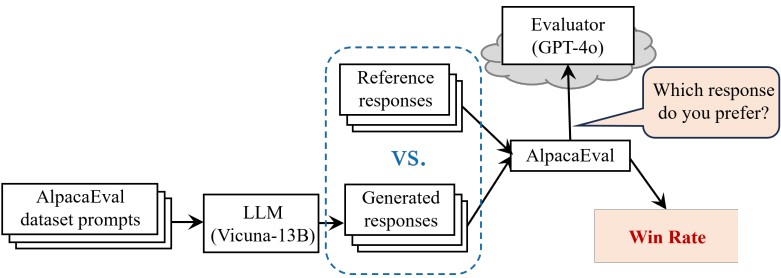

Figure 11: AlpacaEval evaluation pipeline.

while a very high true positive rate (TPR) is achieved for all but the PAIR attacks, the false positive rate (FPR) is also quite high.

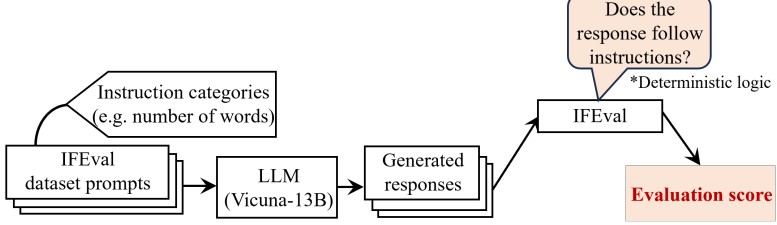

Figure 12: IFEval evaluation pipeline.

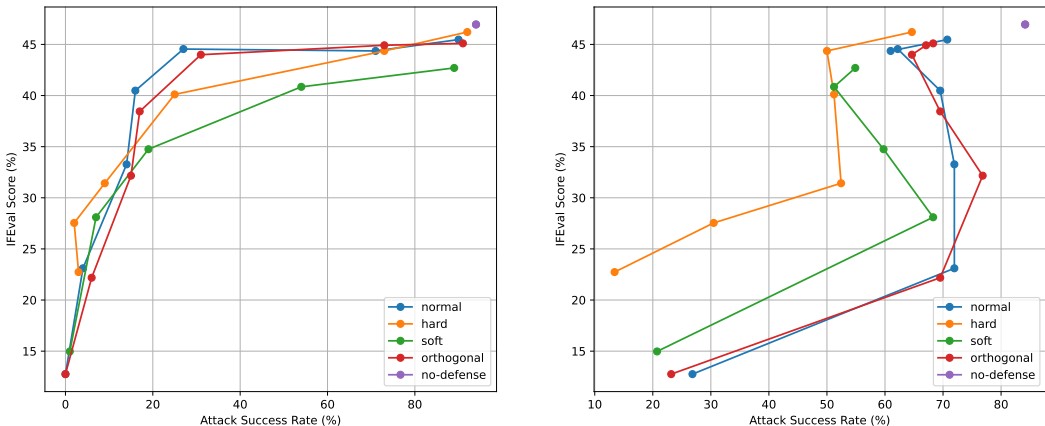

Figure 13: RESTA Performance Tradeoff: Robustness (GCG ASR) vs Utility (IFEval) for Vicuna

Figure 14: RESTA Performance Tradeoff: Robustness (PAIR ASR) vs Utility (IFEval) for Vicuna

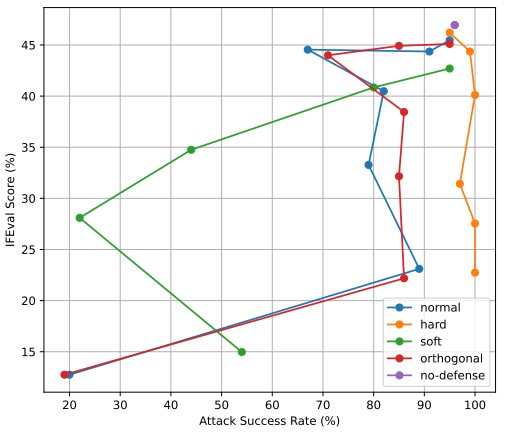

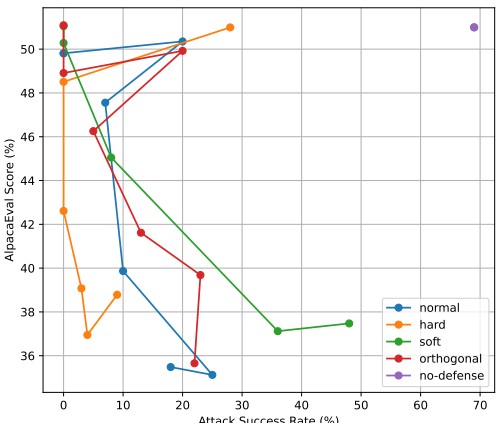

Figure 15: RESTA Performance Tradeoff: Robustness (RS ASR) vs Utility (IFEval) for Vicuna

Figure 16: RESTA Performance Tradeoff: Robustness (RS ASR) vs Utility (AlpacaEval) for Llama

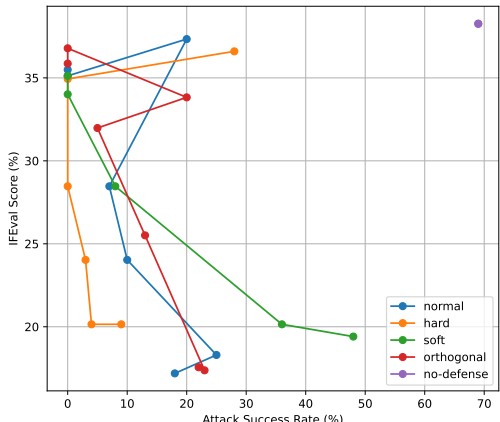

Figure 17: RESTA Performance Tradeoff: Robustness (RS ASR) vs Utility (IFEval) for Llama

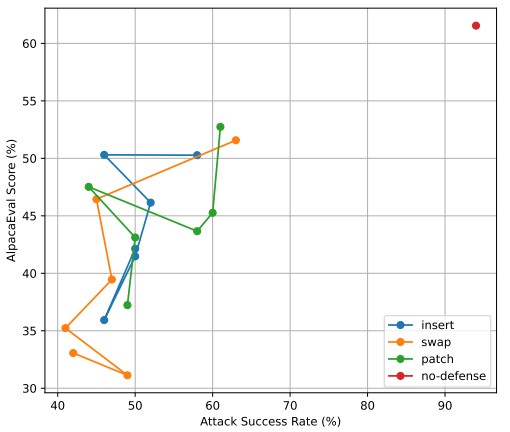

Figure 18: Character Perturbation Performance Tradeoff: Robustness (GCG ASR) vs Utility (AlpacaEval) for Vicuna

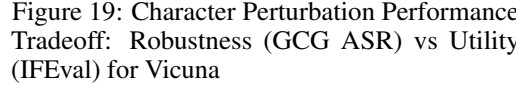

Figure 19: Character Perturbation Performance Tradeoff: Robustness (GCG ASR) vs Utility (IFEval) for Vicuna

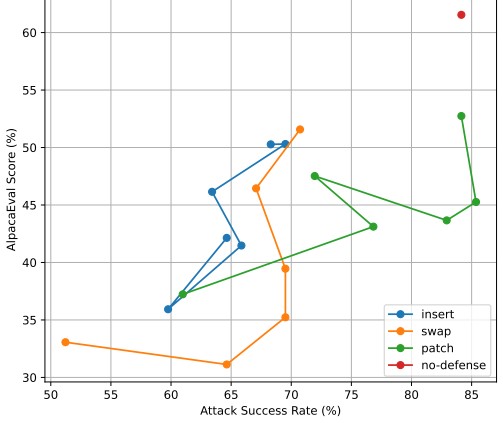

Figure 20: Character Perturbation Performance Tradeoff: Robustness (PAIR ASR) vs Utility (AlpacaEval) for Vicuna

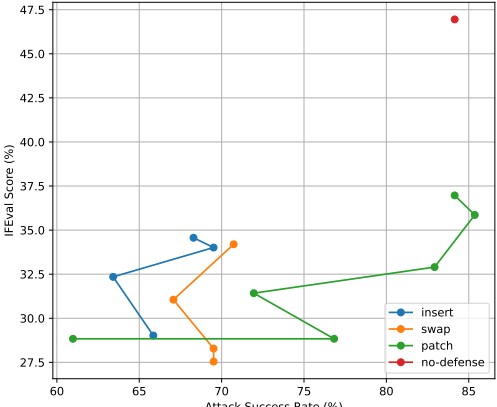

Figure 21: Character Perturbation Performance Tradeoff: Robustness (PAIR ASR) vs Utility (IFEval) for Vicuna

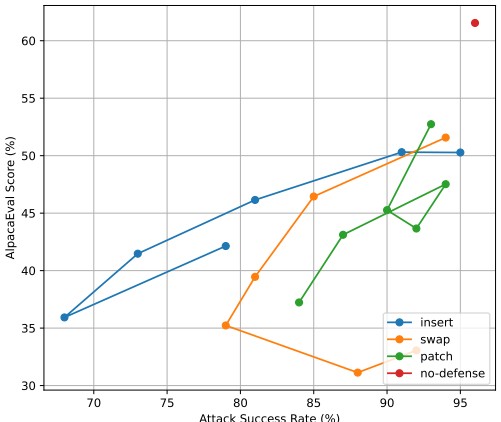

Figure 22: Character Perturbation Performance Tradeoff: Robustness (RS ASR) vs Utility (AlpacaEval) for Vicuna

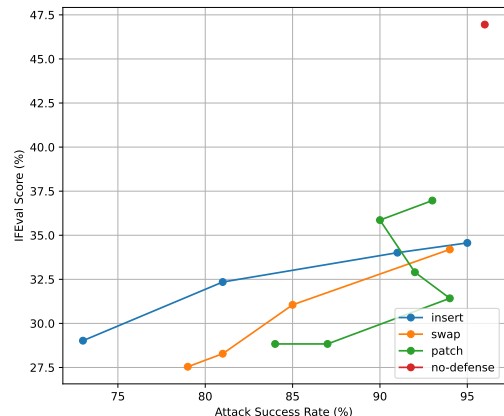

Figure 23: Character Perturbation Performance Tradeoff: Robustness (RS ASR) vs Utility (IFEval) for Vicuna

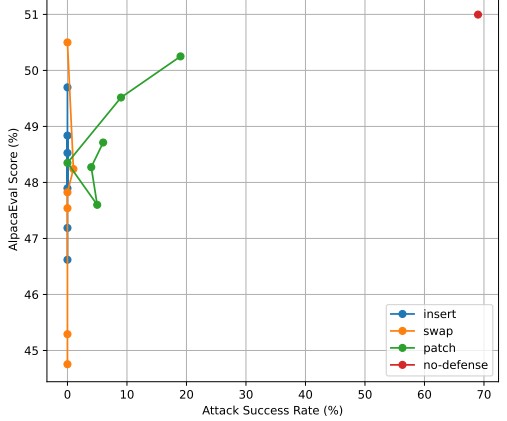

Figure 24: Character Perturbation Performance Tradeoff: Robustness (RS ASR) vs Utility (AlpacaEval) for Llama

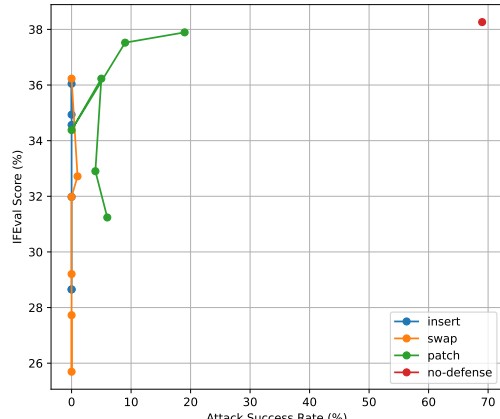

Figure 25: Character Perturbation Performance Tradeoff: Robustness (RS ASR) vs Utility (IFEval) for Llama

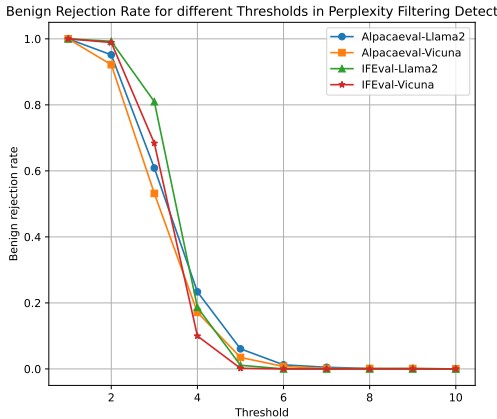

Figure 26: Benign Prompt Rejection Rate vs Threshold for Perplexity Filtering Defense.

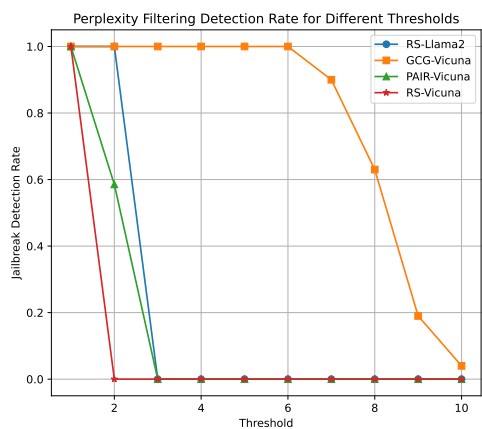

Figure 27: Jailbreak Prompt Detection Rate vs Threshold for Perplexity Filtering Defense

