# OpenReview forum: "Smoothed Embeddings for Robust Language Models"
_NeurIPS.cc/2024/Workshop/SafeGenAi — SafeGenAi Poster_

### Official Review · Reviewer_ywTP · 2024-10-09
**Not very new, but still a contribution**

**Rating:** 6
**Confidence:** 4

**Review:**

### Summary
The paper presents the Randomized Embedding Smoothing and Token Aggregation (RESTA) defense to enhance the robustness of large language models (LLMs) against adversarial attacks, specifically jailbreaking attacks. RESTA operates by adding noise to embedding vectors and aggregating tokens during the generation process. The primary focus is on preserving semantic information while disrupting adversarial perturbations, aiming to achieve a superior robustness versus utility tradeoff compared to other methods. The authors evaluated their method using Vicuna-13B and Llama-2-7B models against multiple attack techniques and observed significant improvements in robustness while maintaining model utility.

### Strengths
1. **Application-Specific Contribution:** While the core method of randomized smoothing is not entirely new, the adaptation of this technique to large language models in the context of adversarial defense is a unique contribution. RESTA's focus on token aggregation and embedding smoothing in this domain adds a novel aspect to its application.
2. **Token-Level Aggregation:** The approach's token-level aggregation method enhances robustness by reducing the impact of any single adversarial token, which is a unique strategy among existing defenses.
3. **Efficiency:** The use of response prefix smoothing minimizes computational costs by limiting the defense to the first few tokens of the generated sequence, making it computationally efficient.
4. **Comprehensive Evaluation:** The experiments are thorough, with evaluations conducted on multiple datasets and attack scenarios, demonstrating the method's robustness and utility tradeoffs.

### Weaknesses
1. **Lack of Baseline Discussion:** Although RESTA is compared with SmoothLLM, the analysis lacks a detailed discussion on other state-of-the-art defenses and how RESTA directly competes with them.
2. **Noise Impact on Semantics:** While different types of noise are explored, there is limited insight into how these perturbations affect the underlying semantics of the embeddings, which could influence the model's ability to generate coherent responses.
3. **Computational Complexity:** Although response prefix smoothing reduces computational costs, the method still requires running multiple noisy samples in parallel, which may not be feasible for real-time applications.
4. **Limited Theoretical Exploration:** The theoretical discussion in the paper is relatively limited, and could be significantly strengthened by referencing the approaches and analyses like https://arxiv.org/abs/2202.01186.

---

### Official Review · Reviewer_SNi9 · 2024-10-10
**This paper introduces Randomized Embedding Smoothing and Token Aggregation (RESTA), a simple but effective model-agnostic defense mechanism**

**Rating:** 8
**Confidence:** 5

**Review:**

This paper introduces Randomized Embedding Smoothing and Token Aggregation (RESTA), a defense mechanism that combats jailbreaking attacks on large language models (LLMs). The proposed approach defends by adding noise to embedding vectors and aggregating.

The topic is particularly compelling given the critical importance of safeguarding LLMs. The simplicity of the method is balanced by its potential effectiveness, as RESTA is both model-agnostic and unsupervised, making it applicable to a wide range of models without requiring additional training process.

The experimental evaluation is thorough and robust. In particular, this work not only assesses the defense effectiveness but also considers the utility of the model, recognizing that introducing noise into the embeddings may impact the normal behavior of LLMs. This dual focus on both security and model utility strengthens the credibility of the approach.

---

### Official Review · Reviewer_6uFn · 2024-10-11
**Overall good paper, accept**

**Rating:** 7
**Confidence:** 4

**Review:**

**Summary:** The paper introduces Randomized Embedding Smoothing and Token Aggregation (RESTA), a defense mechanism for large language models (LLMs) aimed at improving robustness against adversarial attacks such as jailbreaking. RESTA applies noise to the embedding vectors and aggregates token generation results across multiple noisy samples, disrupting adversarial perturbations while preserving semantic information. The approach is tested on Vicuna-13B and Llama-2-7B models using various attack scenarios from JailbreakBench and is compared to other defenses like SmoothLLM. Experimental results show that RESTA reduces attack success rates (ASR) while maintaining utility, particularly in comparison to character-level perturbation methods.

**Strengths / Weaknesses:** The paper shows strong empirical results and a significant increase towards robustness at minimal cost. It would be interesting to see evaluation on more recent models, such as Llama-3, however.

**Correctness:** The results and evaluation methods seem to be valid and align with similar work.

**Clarity:** The paper is quite clear and well-written